# From Clinical Specimen to Whole Genome Sequencing of A(H3N2) Influenza Viruses: A Fast and Reliable High-Throughput Protocol

**DOI:** 10.3390/vaccines10081359

**Published:** 2022-08-19

**Authors:** Cristina Galli, Erika Ebranati, Laura Pellegrinelli, Martina Airoldi, Carla Veo, Carla Della Ventura, Arlinda Seiti, Sandro Binda, Massimo Galli, Gianguglielmo Zehender, Elena Pariani

**Affiliations:** 1Department of Biomedical Sciences for Health, University of Milan, 20133 Milan, Italy; 2Department of Biomedical and Clinical Sciences “L. Sacco”, University of Milan, 20157 Milan, Italy; 3CRC-Coordinated Research Center “EpiSoMI”, University of Milan, 20122 Milan, Italy; 4Interuniversity Research Center on Influenza and Other Transmissible Infections (CIRI-IT), University of Genoa, 16132 Genoa, Italy

**Keywords:** influenza A(H3N2) virus, whole genome sequencing (WGS), next-generation sequencing (NGS), clinical specimen, molecular surveillance

## Abstract

(1) Background: Over the last few years, there has been growing interest in the whole genome sequencing (WGS) of rapidly mutating pathogens, such as influenza viruses (IVs), which has led us to carry out in-depth studies on viral evolution in both research and diagnostic settings. We aimed at describing and determining the validity of a WGS protocol that can obtain the complete genome sequence of A(H3N2) IVs directly from clinical specimens. (2) Methods: RNA was extracted from 80 A(H3N2)-positive respiratory specimens. A one-step RT-PCR assay, based on the use of a single set of specific primers, was used to retro-transcribe and amplify the entire IV type A genome in a single reaction, thus avoiding additional enrichment approaches and host genome removal treatments. Purified DNA was quantified; genomic libraries were prepared and sequenced by using Illumina MiSeq platform. The obtained reads were evaluated for sequence quality and read-pair length. (3) Results: All of the study specimens were successfully amplified, and the purified DNA concentration proved to be suitable for NGS (at least 0.2 ng/µL). An acceptable coverage depth for all eight genes of influenza A(H3N2) virus was obtained for 90% (72/80) of the clinical samples with viral loads >10^5^ genome copies/mL. The mean depth of sequencing ranged from 105 to 200 reads per position, with the majority of the mean depth values being above 103 reads per position. The total turnaround time per set of 20 samples was four working days, including sequence analysis. (4) Conclusions: This fast and reliable high-throughput sequencing protocol should be used for influenza surveillance and outbreak investigation.

## 1. Introduction

Molecular characterization and phylogenetic analysis of viral genomes are critical biomolecular tools in epidemiology, clinical virology, and public health. Genomic sequencing has evolved rapidly in the last years. Sanger sequencing has been the standard method in DNA sequencing for almost 40 years. However, next-generation sequencing (NGS) methodologies are now revolutionizing this branch of science, including epidemiology and virology [1,2]. The experience of the recent SARS-CoV-2 pandemic has further highlighted the need to have NGS workflows promptly available in order to expand public health preparedness and response capacity: in fact, NGS contributed to determine SARS-CoV-2 ecology and spread pattern, and it proved to be an unbiased way to accurately identify new virus variants and study their evolution, which are critical aspects to design effective control and prevention strategies [3,4]. 

NGS techniques provide an enormous volume of sequences in a short time offering new opportunities and insights in research and diagnostic settings [1,5,6]. Many NGS platforms are now available with differences in reaction biochemistry and in sequencing protocol [7]. Numerous genomes of viruses, including influenza viruses (IVs), are currently investigated using high-throughput sequencing methodologies since they provide new opportunities to study viral diversity and evolution, such as to find out multiple drug resistant or immune escape strains, to detect the distribution of viral variants, and to investigate the transmission cluster [1,5,8]. Moreover, while the Sanger sequencing method only enables us to identify the dominant variant in a sample, using deep sequencing technology it is possible to detect rare genomic variants within genetically diverse viral populations [9,10]. Moreover, Sanger sequencing requires prior knowledge of the genomic sequences under study, while NGS technologies can be used for both target-dependent and target-independent nucleic acid sequencing, thus playing important roles in the identification and characterization of novel pathogens [11,12,13]. 

IVs (*Orthomyxoviridae* family) are single-stranded, negative sense, segmented RNA viruses. To date, IVs are divided into four types (A–D) according to the antigenic differences of the viral nucleoprotein and matrix proteins [14,15]. IVs type A (IAV) and B (IBV) are further classified into subtypes or lineages, respectively, on the basis of serotype-specific antigenic determinants of their two surface glycoproteins, the haemagglutinin (HA) and the neuroaminidase (NA) [16,17]. Nowadays, among IAVs, 18 variants of HA and 11 variants of NA have been identified, enabling their classification into numerous subtypes [17,18]. A(H3N2) and A(H1N1) are currently the two main IAV subtypes circulating in human population along with the two IBV lineages (B-Yamagata and B-Victoria) [19,20]. 

IVs, and IAVs particularly, are constantly evolving through two different ways, the so-called antigenic drifts and antigenic shifts; they change their genetic and antigenic determinants and promote the emergence of new viral variants that can cause epidemics and even pandemics [21]. A(H3N2) has a higher mutation rate than A(H1N1) and has undergone frequent antigenic drifts since its emergence in 1968 [22]. The high genetic and antigenic variability of the viral surface proteins and consequently the simultaneous circulation of different IV subtypes make the burden of the disease extremely unpredictable [23,24,25]. As emphasized by the World Health Organization (WHO) through the Global Influenza Surveillance and Response System (GISRS), it is essential to monitor circulating IVs in order to detect viral genetic changes and study viral evolution. The GISRS safeguards public health by providing various recommendations regarding laboratory diagnostics, vaccines, antiviral susceptibility, and risk assessment [26]. Traditionally, virological surveillance has been conducted through the partial genome sequencing of IVs, focusing mainly on HA and NA genes, using the Sanger method. In order to obtain sufficient starting material for sequencing, the viral strains are sometimes previously grown in chicken eggs or propagated in cell cultures [27]. As encouraged by the Centers for Disease Control and Prevention’s (CDC) Advanced Molecular Detection (AMD) initiative, using clinical samples combined with an advanced sequencing method, such as NGS technology, could be less time-consuming and achieve a deep molecular analysis of IVs [28].

We aimed at describing an accurate, time-saving and cost-effective molecular protocol for carrying out whole genome sequencing (WGS) of A(H3N2) IVs directly from clinical respiratory specimens, to be used for influenza surveillance.

## 2. Experimental Design

As the regional reference laboratory of the epidemiological and virological influenza surveillance in the framework of the Italian Influenza Surveillance Network (InfluNet), we analyzed 549 respiratory specimens collected from outpatients with influenza-like illness (525/549; 96%) or from inpatients with severe respiratory infections (24/549; 4%) in Lombardy (Northern Italy) during the 2016/2017 influenza season [29]. Data and samples from patients with mild and severe influenza were collected and analyzed anonymously within the National Influenza Surveillance Program, and they were managed according to the Good Laboratory Practice procedures. 

The clinical samples were oropharyngeal (OP) swabs or broncho-alveolar lavages (BAL), which were analyzed to detect IVs using specific one-step real-time PCR assays in accordance with international standardized protocols [27]. Hence, 249 out of 549 samples (45%) tested positive to A(H3N2) IV and their viral load was quantified by a real-time RT-PCR [30]; 80 A(H3N2)-positive samples (80/249; 32%) with viral loads ranging from 10^4^ to 10^10^ genome copies/mL were included in this study. Demographical and virological data of study population are summarized in Appendix A.

RNA was extracted from respiratory specimens and a one-step RT-PCR assay was used to retro-transcribe and amplify the entire IAV genome in a single reaction. No DNase treatment to remove human genome was needed since the retro-transcription and the amplification were performed by using specific primers targeting directly the viral RNA. The PCR amplicons were cleaned-up and purified DNA was quantified. Genomic libraries were prepared and sequenced by using Illumina MiSeq platform. The obtained reads were evaluated for sequence quality and assembled as consensus.

The total turnaround time of the processing of about 20 clinical specimens to their sequence analysis requires up to a maximum of four working days (including sequence analysis), as shown in the workflow in Figure 1. 

## 3. Materials and Equipment

### 3.1. Clinical Specimens

OP swabs were collected by means of Sigma Virocult^®^ kit (Medical Wire—MWE, Corsham, UK) that consisted in a prepackaged sterile kit containing a small vial with 1 mL of viral transport medium (VTM), stable at room temperature, and a dry swab composed by an open-celled foam bud and a stick with a specific breakpoint [31].

BAL were collected through a minimally invasive procedure that involves instillation of sterile normal saline (facilitated by the introduction of a flexible bronchoscope) into a sub-segment of the lung, followed by suction and collection of the instillation for analysis.

All samples were collected by skilled healthcare personnel. Once collected, clinical samples were stored at +4 °C and sent to the laboratory within 72 h for virological analyses. For the protocol here proposed, we used residual volume of samples stored at −80 °C.

### 3.2. Nucleic Acid Extraction


*Reagents*


-Invisorb^®^ Spin Virus RNA Mini kit (Invitek Molecular GmbH, Berlin, Germany) for 250 viral RNA extractions, consisted of:
Lysis Buffer RV, ready-to-uselyophilized Proteinase K, resuspended by adding 2 mL ddH_2_O to each tubelyophilized Carrier RNA, resuspended by adding 2 mL RNase free water (not provided) to each tubeBinding Solution, obtained by filling 120 mL 99.7% isopropanol into the empty bottle at the first use.concentrate Wash Buffer R1, diluted by adding 80 mL 96–100% ethanol to each bottle at the first use.concentrate Wash Buffer R2, diluted by adding 160 mL 96–100% ethanol to each bottle at the first use.Elution Buffer, ready-to-useRTA spin filter2 mL RTA receiver tubes2 mL safe-lock tubes1.5 mL elution microcentrifuge tubes


*Equipment*


-vortex mixer-thermomixer-high speed benchtop microcentrifuge


*Protocol*


LYSIS PHASE

(1)Pipet 600 µL of Lysis Buffer RV into a 2 mL safe-lock tube.(2)Transfer 20 µL of Proteinase K and 20 µL of Carrier RNA into the 2 mL safe-lock tube containing the lysis buffer.(3)Mix by pulse-vortexing the clinical sample and then add 200 µL of specimen to the lysis buffer-proteinase K-carrier RNA in the 2 mL safe-lock tube.(4)Mix by pulse-vortexing the lysis solution for 15 s and briefly centrifuge the tube to remove drops from the inside of the lid.(5)Incubate the tube into a thermomixer at 65 °C for 10 min.(6)Briefly centrifuge the tube to remove drops from the inside of the lid.

RNA BINDING PHASE

(7)Add 400 μL Binding Solution to the lysate sample, and mix by pulse-vortexing for 20 s. After mixing, briefly centrifuge the tube to remove drops from inside the lid.(8)Carefully apply 650 μL of the solution from step 7 to the RTA spin filter (in a 2 mL RTA receiver tube) without wetting the rim. Close the cap, and incubate at room temperature for 1 min. Then, centrifuge at 8000 rpm for 1 min. Discard the filtrate and place the RTA spin filter back into its 2 mL RTA receiver tube.(9)Repeat step 8 to load onto the spin column the left volume (approximately 600 μL) of the lysate. Place the RTA spin filter into a clean 2 mL RTA receiver tube.

WASHING PHASE

(10)Add 600 μL Wash Buffer R1 into the RTA spin filter, close the cap, and centrifuge at 8000 rpm for 1 min. Place the RTA spin filter in a clean 2 mL RTA receiver tube, and discard the tube containing the filtrate.(11)Add 600 μL Wash Buffer R2 into the RTA spin filter, close the cap, and centrifuge at 8000 rpm for 1 min. Place the RTA spin filter in a clean 2 mL RTA receiver tube, and discard the tube containing the filtrate.(12)Repeat the washing step 11.(13)To eliminate possible Wash Buffer R2 carryover, centrifuge the RTA spin filter at full speed (around 12,000 rpm) for 4 min.

ELUTION PHASE

(14)Place the RTA spin filter in a clean 1.5 mL elution microcentrifuge tube. Discard the old RTA receiver tube containing the filtrate. Carefully, add 100 μL Elution Buffer (previously heated at 80 °C) directly onto the RTA spin filter and incubate at room temperature for 3 min.(15)Centrifuge the microcentrifuge tube with the RTA spin filter at 8000 rpm for 1 min. Discard the RTA spin filter and store the microcentrifuge tube with the eluate for further bio-molecular analyses.

### 3.3. Viral Genome Amplification


*Reagents*


-Super Script™ III One-Step RT-PCR System with Platinum™ Taq DNA Polymerase (ThermoFisher Scientific, Waltham, MA, USA), consisted of:
2X Reaction Mix (with MgSO_4_ included at a final concentration of 1.6 mM)SuperScript™ III RT/Platinum™ Taq Mix-MBTuni-12 forward primer and MBTuni-13 reverse primer (working concentration 30 µM) [32]-DNase/RNase-free water (Sigma-Aldrich, St. Louis, MO, USA)


*Equipment*


-vortex mixer-benchtop microcentrifuge-thermal-cycler


*Protocol*


(1)Gently mix the 2X Reaction Mix, the SuperScript™ III RT/Platinum™ Taq Mix and the primer tubes.(2)In a sterile, nuclease-free microcentrifuge tube, combine the following components on ice: 25 µL of 2X Reaction Mix Buffer, 2 µL of forward (MBTuni-12) and reverse (MBTuni-13) primer, 2 µL of SuperScript™ III RT/Platinum™ Taq Mix and 14 µL of DNase/RNase-free water to reach a final volume of 45 µL.(3)Add 15 µL of extracted RNA to the reaction mixture.(4)Mix by vortexing and centrifuge briefly.(5)Place the reaction tube into a thermal-cycler programmed as follow: 42 °C for 1 h for the RT phase followed by 94 °C for 2 min for enzyme inactivation; 5 cycles at 94 °C for 30 s, 45 °C for 30 s and 68 °C for 3 min, 35 cycles at 94 °C for 30 s, 57 °C for 30 s, 68 °C for 3 min, followed by 1 cycle at 68 °C for 7 min.

### 3.4. Amplicon Purification


*Reagents*


-NucleoSpin Gel^®^ and PCR Clean-up kit (Macherey-Nagel, Düren, Germany) for 250 preparations, consisted of:
Buffer NTI, ready-to-useconcentrated Buffer NT3, diluted by adding 200 mL 96–100% ethanol to each bottle at the first use.Buffer NE, ready-to-useNucleoSpin Gel^®^ and PCR Clean-up Column2 mL collection tube


*Equipment*


-dry block heating-high speed benchtop microcentrifuge


*Protocol*


(1)Mix 50 µL sample with 100 µL Buffer NTI (proportion 1:2) by pipetting up and down.(2)Load the solution (150 µL) on the NucleoSpin Gel^®^ and PCR Clean-up Colum. Centrifuge the column for 30 s at 11,000× *g*. Discard flow-through and place the column back into the collection tube.(3)Add 650 µL Buffer NT3 to the NucleoSpin Gel^®^ and PCR Clean-up Column. Centrifuge the column for 30 s at 11,000× *g*. Discard flow-through and place the column back into the collection tube.(4)Repeat the washing step 3.(5)Centrifuge the NucleoSpin Gel^®^ and PCR Clean-up Column for 5 min at 12,000 rpm to remove Buffer NT3 completely.(6)Incubate the NucleoSpin Gel^®^ and PCR Clean-up Column, with the lid opened, for 10 min at room temperature to remove any residual ethanol from Buffer NT3.(7)Place the NucleoSpin Gel^®^ and PCR Clean-up Column into a new 1.5 mL microcentrifuge tube (not provided) and add 20 µL Buffer NE (previously heated at 70 °C). Incubate at room temperature for 1 min. Centrifuge for 5 min at 11,000× *g*.(8)Repeat twice the elution step 7 to have a final elution volume of 60 µL.(9)Discard the NucleoSpin Gel^®^ and PCR Clean-up Column and store the microcentrifuge tube with the eluate for further bio-molecular analyses.

### 3.5. Library Preparation for Whole Genome Sequencing Using NGS Technology


*Reagents*


-Nextera-XT DNA sample preparation kit (Illumina Inc., San Diego, CA, USA) for 96 samples, consisted of:
Amplicon Tagment Mix (ATM)Tagment DNA Buffer (TD)Nextera PCR Master Mix (NPM)Resuspension Buffer (RSB)Library Normalization Additives (LNA1)Library Normalization Wash (LNW1)Hybridization Buffer (HT1)Neutralize Tagment Buffer (NT)Library Normalization Beads 1 (LNB1)Library Normalization Storage Buffer 1 (LNS1)-Nextera-XT DNA sample preparation index kit (Illumina Inc., San Diego, CA, USA) for 96 samples, consisted of tubes with index primers 1 and index primers 2.-Agencourt AMPure XP 60 mL kit (Beckman Coulter Life Sciences, Brea, CA, USA), consisted of AMPure XP Beads.-80% ethanol-0.1 N of NaOH


*Equipment*


-vortex mixer-shaker-96-plate magnetic stand-96-plate benchtop centrifuge-benchtop microcentrifuge-thermal-cycler-dry block heater


*Protocol*


DNA TAGMENTATION

(1)Mix gently and centrifuge briefly the reagents ATM and TD.(2)Transfer 10 µL TD to each well of a 96-plate (plate A) according to the number of samples to be processed.(3)Add 5 µL DNA and 5 µL ATM in each well and mix by pipetting up and down.(4)Seal and centrifuge the plate A for 1 min at 1200 rpm.(5)Place the plate A into a thermal-cycler programmed as follows: 55 °C for 5 min and hold at 10 °C.(6)Remove the plate A from the thermal-cycler and add 5 µL NT in each well; mix the solution by pipetting up and down.(7)Seal and centrifuge the plate A for 1 min at 1200 rpm.(8)Incubate the plate A for 5 min at room temperature.

PCR AMPLIFICATION

(1)Mix gently and centrifuge briefly the reagent NPM and the index primers.(2)Transfer 15 µL NPM in each well of the plate A according to the number of samples to be processed.(3)Add 5 µL of a specific index primer 1 and 5 µL of a specific index primer 2 in each well; mix by pipetting up and down.(4)Seal and centrifuge the plate A for 1 min at 1200 rpm.(5)Place the plate A into a thermal-cycler programmed as follows: 1 cycle at 72 °C for 3 min, followed by 95 °C for 30 s; 12 cycles at 95 °C for 10 s, 55 °C for 30 s and 72 °C for 3 s, followed by 1 cycle at 72 °C for 5 min and hold at 10 °C.

AMPLICON PURIFICATION

(1)Remove the plate A from the thermal-cycler and transfer 50 µL of each well of the plate A to a well of a new plate (plate B).(2)Add 30 µL of AMPure XP Beads in each well.(3)Seal and shake the plate B for 2 min at 1800 rpm; then, incubate for 5 min at room temperature.(4)Place the plate B to a magnetic stand and wait until the liquid is clear (approximately 2 min).(5)Keep the plate B on magnetic stand and remove the supernatant.(6)Add 200 µL of 80% ethanol in each well and incubate for 30 s at room temperature.(7)Keep the plate B on magnetic stand and remove the ethanol.(8)Repeat step 6 and 7.(9)Keep the plate B on magnetic stand and wait 15 min to eliminate residual ethanol.(10)Remove the plate B from the magnetic stand and add 52.5 µL RSB in each well.(11)Seal and shake the plate B for 2 min at 1800 rpm. Then, incubate for 2 min at room temperature.(12)Place the plate B to the magnetic stand and wait until the liquid is clear (approximately 2 min).(13)Transfer 50 µL the supernatant of each well of the plate B to a well of a new plate (plate C).

LIBRARY NORMALISATION

(1)Mix by vortexing the reagents LNA1 and LNB1.(2)In a 50 mL tube, transfer 45.8 µL LNA1 per number of samples and add 8.3 µL LNB1 per number of samples. Invert the tube to mix the solution.(3)Transfer 20 µL of the liquid of each well of plate C to a well of a new plate (plate D).(4)Add 45 µL LNA1/LNB1 solution in each well of the plate D according to the number of samples to be processed.(5)Seal and shake the plate D for 30 min at 1800 rpm.(6)Place the plate D to the magnetic stand and wait until the liquid is clear (approximately 2 min).(7)Keep the plate D on magnetic stand and remove the supernatant.(8)Remove the plate D from the magnetic stand and add 45 µL LNW1 in each well.(9)Seal and shake the plate D for 5 min at 1800 rpm.(10)Place the plate D to the magnetic stand and wait until the liquid is clear (approximately 2 min).(11)Keep the plate D on magnetic stand and remove the supernatant.(12)Repeat steps 8–11.(13)Remove the plate D from the magnetic stand and add 30 µL NaOH 0.1 N in each well.(14)Seal and shake the plate D for 5 min at 1800 rpm.(15)Place the plate D to the magnetic stand and wait until the liquid is clear (approximately 2 min).(16)Keep the plate D on the magnetic stand and transfer 30 µL of the supernatant of each well to a well of a new plate (plate E).(17)Add 30 µL LNS1 into each well of the plate E according to the number of samples to be processed.(18)Seal the plate E and centrifuge for 1 min at 1000× *g*.

SEQUENCING PLATFORM LOADING

(1)Transfer 5 µL of each library into a 1.5 mL microcentrifuge tube (tube A). Then, transfer 24 µL of the pooled library (tube A) into a new 1.5 mL microcentrifuge tube (tube B).(2)Add 576 µL HT1 into the tube B and mix by pipetting up and down.(3)Vortex the tube B and incubate for 2 min at 96 °C.(4)Invert twice the tube B and cool on ice for 5 min.(5)Load the library of the tube B into the MiSeq Reagent Cartridge.(6)Place the MiSeq Reagent Cartridge into the Load Sample reservoir and start the run by using the Illumina MiSeq Instrument (Illumina Inc., San Diego, CA, USA).

## 4. Detailed Procedure

RNA was extracted from 200 μL of each respiratory sample (VTM of OP swabs or BAL) and eluted in 100 μL of elution buffer using the Invisorb^®^ Spin Virus RNA Mini kit (Invitek Molecular GmbH, Berlin, Germany), following the manufacturer’s instructions described step-by-step in Section 3.2.

Then, 15 µL of extracted RNA was used to retro-transcribe and amplify the entire IAV genome (8 RNA fragments) in a single reaction by using the one-step RT-PCR assay detailed in Section 3.3. To verify the amplification results, the 8 segments of amplified IV can be separated on a 1.5% agarose gel, running the electrophoresis at 130 V for approximately one hour against 1Kb DNA ladder (SHARPMASS^TM^, EuroClone, Pero, Milan, Italy).

Each amplified sample was purified by using the commercial kit NucleoSpin Gel^®^ and PCR Clean-up kit (Macherey-Nagel, Düren, Germany) following the modified PCR clean-up protocol, detailed in Section 3.4. Briefly, 50 µL of each amplified sample were eluted in a final volume of 60 µL by means of 3 elution steps with 20 µL of pre-heated elution buffer each in order to increase the recovery of larger fragments (>1000 bp). 

Purified DNA was quantified using a microplate reader (Tecan Group AG, Männedorf, Switzerland). Each sample was diluted to an initial concentration of 0.2 ng/µL in accordance with the Illumina protocol, and 1 ng was used for the library preparation. DNA libraries were made with a Nextera-XT DNA sample preparation kit (Illumina Inc., San Diego, CA, USA) according to the manufacturer’s instructions detailed in Section 3.5. 

Pooled genomic libraries were sequenced on the Illumina MiSeq Instrument (Illumina Inc., San Diego, CA, USA) using the Illumina MiSeq Reagent Kit v2—300 cycles (Illumina Inc., San Diego, CA, USA) in order to generate 151 paired-end reads.

FASTQ files were generated using MiSeq Reporter (Illumina Inc., San Diego, CA, USA) and the paired reads were imported to Geneious software v. R11 (Biomatters, Auckland, New Zealand; available at: https://www.geneious.com/, accessed on 10 August 2022) for read assembly and assembly sequence analysis by using the “map to reference” approach and the 2016/2017 A(H3N2) vaccine virus A/Hong Kong/4801/2014 (retrieved from GISAID—Global Initiative on Sharing All Influenza Data—database, accession number: EPI ISL 198222) as reference strain. The analysis was performed following the Geneious Manual, section “Assembly and Mapping”, paragraph “Map to reference” [33]. For thoroughness, other bioinformatics software can be used as long as all these steps are followed: (a) the reads are to be paired, (b) all paired read data have to be of known expected distance between each pair (length: 151 nt, in our case), (c) the paired reads have to be aligned to the reference strain A/Hong Kong/4801/2014 (GISAID accession number: EPI ISL 198222) and (d) the aligned reads have to be assembled in a consensus sequence.

The raw data reads with quality values (QV) >20 were filtered by excluding contaminants, such as adapters, the ambiguous “N” nucleotides, and low-quality sequences, using trimming options available in the bioinformatics software.

## 5. Results

All the study samples were successfully amplified to obtain the entire IAV genome. All purified DNA was of a suitable quality and concentration (at least 0.2 ng/µL) to be sequenced by NGS.

Overall, an acceptable coverage depth for all eight fragments of IAV was obtained for clinical samples with viral loads >10^5^ genome copies/mL (72/80, 90%). The mean coverage depth for each viral genome segment is shown in Table 1, ranging from 1101.45 for the polymerase basic 1 (PB1) gene to 10,475.41 for the matrix 1 (M1) gene.

In these clinical samples, mapping to the reads on the reference sequence allowed extracting the number of times each position was covered, i.e., the depth of sequencing. The mean depth of sequencing ranged from 200 to 105 reads per position, with the majority of the mean depth values being above 103 reads per position. The mean number of mapped bases and reads for each IAV gene is detailed in Table 1. The lowest values were obtained for PB1 and polymerase basic 2 (PB2) genes (2,596,691.75 for PB1 and 33,164.68 for PB2, respectively), while the highest ones for M1 gene (10,126,540.35 and 73,918.02, respectively). All the study samples were of comparable quality and no outliers were observed.

## 6. Discussion

We aimed at describing and investigating the validity of a WGS protocol to obtain the complete genome of A(H3N2) IVs directly from clinical specimens by analyzing 80 A(H3N2)-positive respiratory samples. 

Over the last ten years, there has been growing interest in WGS of rapidly mutating pathogens, such as IVs, with the aim of enhancing biological and medical knowledge [1,5,6,8]. However, this molecular approach requires simple, rapid, and reliable protocols to use in diagnostic and clinical research settings. The bio-molecular protocol used in this study in order to obtain WGS data on A(H3N2) IVs is based on the combination of three fundamental characteristics which can be advantageous for influenza surveillance activities. Firstly, using clinical samples provides us with a cell culture-independent protocol to sequence the whole genome of IAVs. Avoiding virus propagation in cell cultures, as instead a previously study did [34], or in chicken eggs [27], is much less time consuming and expensive. Moreover, it circumvents the difficulty of growing A(H3N2) in cell cultures (often observed in recent years [35]) and prevents the emergence of cell- or egg-adaptive mutations in sequenced strains, that could misrepresent the characteristics of circulating viruses [36,37,38]. Using a single set of specific primers (one forward and one reverse primer) to retro-transcribe and amplify all eight segments of the IAV genome by a one-step RT-PCR assay [32] is the second valuable attribute of the protocol. This approach allows an efficient viral target enrichment in a single reaction thus avoiding more laborious and sometimes expensive enrichment steps (such as commercial target enrichment methods [39], ultracentrifugation [40,41], or a series of nested PCR runs [42,43,44]) and additional human genome removal treatments (such as DNAse digestion [40,41,43,45,46]). Moreover, having a specific amplified target to be sequenced using NGS, prevents us from having to filter out reads originated from human genome within the bioinformatics analyses [47]. Therefore, this approach improves the quality of sequencing results and reduces time and costs. Moreover, it is well-known that having two consecutive steps in a single tube without the manual intervention of an operator reduces the risk of cross-contamination between biological samples [27,48] and generates better final results. Finally, using an NGS platform to obtain the WGS of IAVs guarantees a rapid and thorough sequencing. 

The recent advances made in NGS techniques have revolutionized their application in several fields, and have increased speed, read length and throughput and greatly reduced per-base costs. The main advantage of NGS methodologies is that they provide an enormous volume of sequences in short time at relatively low costs and can be used for a broad range of applications in research and diagnostic settings [1,5]. It is important to note that the Illumina MiSeq platform used in this study has proven to be a valuable platform for microbial genome sequencing. It is fast and easy to use and produces accurate sequencing results [49].

Overall, the WGS protocol used in this study has proven to be an efficient and effective approach since 80 clinical samples were analyzed and high-quality sequencing results were obtained for 90% of tested samples for all eight segments of IAV genome, including the largest genes encoding for the polymerases (PB1, PB2, and PA), which previous studies [50,51] failed to obtain.

With a total turnaround time of four working days, including sequence analysis, this has proven to be both fast and easy and can promote more rapid and exhaustive data sharing, which means that it is a suitable protocol for epidemiological and virological surveillance activities and vaccine effectiveness studies. 

The availability of IV whole genomes obtained through NGS-based bio-molecular protocols, may also promote new collaborations among researchers to carry out in-depth studies on molecular characterization and phylogenetic analysis. However, it is important to note that processing a larger number of specimens at the same time (i.e., 20 samples per run as occurred in this study) is much less time-consuming and expensive than testing a single sample per run. However, it should be noted that NGS technologies produce such large amounts of data and therefore require high-performance computing resources and skilled personnel with strong bioinformatics capabilities [1,5]. 

As previously demonstrated, the WGS of IVs is an important approach for influenza surveillance and for studying viral diversity and evolution. It can help us to identify genetic variations among circulating IVs, to promptly recognize the emergence of new reassortant strains, and to analyze in depth the genetic changes that can lead to viral variants of particular interest to public health, including viruses with pandemic potential. WGS is a useful tool for: (i) predicting the gain/loss of potential glycosylated sites, since these post-translational modifications may play an important role in viral transmission, antigenicity, virulence, affinity and receptor binding capacity; (ii) predicting antiviral resistance; and (iii) estimating the genetic homology between epidemic strains and vaccine strains, by evaluating the HA and NA genes as well as others [46,52,53,54,55,56].

## 7. Conclusions

In conclusion, according to the CDC’s AMD program, the availability of new additional information on pathogen genomes at a lower cost and in a short time thanks to the NGS-based biomedical protocol here described is particularly important considering the global impact of influenza on public health. Moreover, this protocol may enable us to obtain more effective vaccines which match the circulating strains and to identify new viruses with high pathogenicity and pandemic potential.

## Figures and Tables

**Figure 1 vaccines-10-01359-f001:**
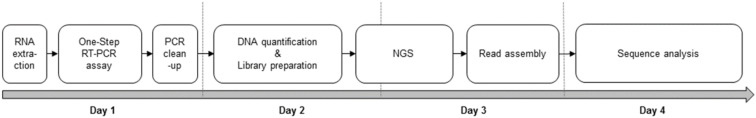
Workflow and turnaround time per set of 20 clinical respiratory specimens.

**Table 1 vaccines-10-01359-t001:** Mean number of mapped bases and reads, and average coverage depth for each IAV gene of study samples with viral loads >105 genome copies/mL (72/80, 90%).

	*Influenza A Virus Genome*							
	Segment 1	Segment 2	Segment 3	Segment 4	Segment 5	Segment 6	Segment 7	Segment 8
	*Polymerase Basic 2 (PB2)*	*Polymerase Basic 1 (PB1)*	*Polymerase Acid (PA)*	*Hemagglutinin (HA)*	*Nucleoprotein (NP)*	*Neuraminidase (NA)*	*Matrix 1* *(M1)*	*Non Structural (NS)*
	2341 bps	2341 bps	2233 bps	1778 bps	1565 bps	1413 bps	1027 bps	890 bps
** *Mean N.* ** ** *of mapped* ** ** *bases* **	4,754,119.20	2,596,691.75	5,418,931.93	6,023,123.21	6,450,089.98	6,612,054.32	10,126,540.35	9,175,172.23
** *Mean N.* ** ** *of mapped reads* **	33,164.68	37,966.80	38,286.33	43,176.47	44,939.80	44,097.74	73,918.02	64,266.64
** *Mean* ** ** *coverage depth* **	2154.15	1101.45	2694.11	3745.52	3521.58	3933.64	10,475.41	8348.54

## Data Availability

The datasets generated during and/or analyzed during the current study are available from the corresponding author upon request.

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
