# Peer review of "From Clinical Specimen to Whole Genome Sequencing of A(H3N2) Influenza Viruses: A Fast and Reliable High-Throughput Protocol"

_vaccines, 2022, doi:10.3390/vaccines10081359_

Round 1
Reviewer 1 Report
The work by Cristina and collaborators(From clinical specimen to whole genome sequencing of A(H3N2) influenza viruses: a fast and reliable high-throughput protocol)used Illumina MiSeq platform to obtain the complete genome sequence of A(H3N2) IVs directly from clinical specimens, and determined the validity of a WGS protocol. A very innovative work was carried out using the latest sequencing technology, that’s good.
However, This manuscript gives me the feeling that it has just ended at the beginning. The experimental design is not complete and the research content is less. For example, the repeatability of the same sample? Authors need more samples and more repetitive work to verify the validation of fast and reliable high-throughput protocol.
Minor Issues:
Many superscripts are not well marked, such as line 33 and 157, which need to be edited
Reviewer 2 Report
The manuscript submitted by Cristina Galli described the method for influenza virus next generation sequencing, which has been proven to be extremely valuable for virus surveillance and molecular epidemiology investigation.
However, the method described in this manuscript has been published previously and has been widely used in many laboratories, therefore the novelty of this study is very limited.
Although the authors claim their method to be fast and reliable, no actual improvements have been done to make the existing protocol run faster and more reliable. For example, for PCR clean-up step, instead of using Spin column protocol as described in this manuscript, purification using AMPure magnetic beads is more efficient and more suitable for high throughput. Additionally, superfast enzyme such as SuperScript IV one-step RT-PCR system can replace Superscript III One-Step RT-PCR system if they want to shorten the time for RT-PCR procedure.
Lastly, they showed that the viral threshold for NGS analysis is approximately 105 genome copies/ml. This threshold seems to be lower than that described in other studies especially considering no DNase treatment after viral RNA extraction was performed in this study.
Since only the mean mapped reads and coverage depth were reported in this study, it is not clear what the mapped reads and coverage depth were for low genome copy samples.
Authors may consider commenting on this in the result section.
Reviewer 3 Report
In this manuscript, the authors present their protocol to sequence influenza virus using MiSeq machine. Although a large number of samples were used, it is not clear how this method is any different from other conventional methods used around the world. Simply put, this study lacks the novelty. More specific comments are listed below:
Major points:
[1] “2.4. Viral genome amplification”. As far as what is written in this subsection, the authors did NOT treat RNA with DNase I to digest the genomic DNA in prior to reverse transcription. Although influenza virus is RNA virus, the rationale for not treating with DNase I must be stated clearly.
[2] “Geneious software 143 v. R11 (Biomatter, New Zealand; available at: https://www.geneious.com/)” The authors used the commercial available software product to analyze the generated data. Unless this product is freely available to the academic community, the authors must provide the computational pipeline using other freely available software products (i.e., without license fees) as it is not fair for others without access to this commercial software product, especially the authors clearly states as follow in the abstract: “The total turnaround 33 time per set of 20 samples was 4 working days, including sequence analysis.”
[3] “The study samples (N=80) had viral loads ranging from 104 to 1010 genome cop-157 ies/ml; all of which were successfully amplified to obtain the entire IAV genome; all puri-158 fied DNA was of a suitable quality and concentration (at least 0.2 ng/μl) to be sequenced 159 by NGS.” The authors must provide a table summarizing the patients’ information, such as age, gender, etc.
Minor points:
(1) The generated RNA-seq data must be deposit in a public domain, such as GEO.
Round 2
Reviewer 1 Report
All of my concerns have been responded. I recommend the publication of this paper
Author Response
We would like to thank you for your revision and support.
Reviewer 2 Report
I agree that reediting the manuscript in a protocol format makes it more suitable for publication and the revised manuscript has fully addressed my comments from initial review.
Author Response

(The authors gave the same response as above.)

Reviewer 3 Report
The point-by-point responses to the previous comments and suggestions by this reviewer are missing:
In this manuscript, the authors present their protocol to sequence influenza virus using MiSeq machine. Although a large number of samples were used, it is not clear how this method is any different from other conventional methods used around the world. Simply put, this study lacks the novelty. More specific comments are listed below:
Major points:
[1] “2.4. Viral genome amplification”. As far as what is written in this subsection, the authors did NOT treat RNA with DNase I to digest the genomic DNA in prior to reverse transcription. Although influenza virus is RNA virus, the rationale for not treating with DNase I must be stated clearly.
[2] “Geneious software 143 v. R11 (Biomatter, New Zealand; available at: https://www.geneious.com/)” The authors used the commercial available software product to analyze the generated data. Unless this product is freely available to the academic community, the authors must provide the computational pipeline using other freely available software products (i.e., without license fees) as it is not fair for others without access to this commercial software product, especially the authors clearly states as follow in the abstract: “The total turnaround 33 time per set of 20 samples was 4 working days, including sequence analysis.”
[3] “The study samples (N=80) had viral loads ranging from 104 to 1010 genome cop-157 ies/ml; all of which were successfully amplified to obtain the entire IAV genome; all puri-158 fied DNA was of a suitable quality and concentration (at least 0.2 ng/μl) to be sequenced 159 by NGS.” The authors must provide a table summarizing the patients’ information, such as age, gender, etc.
Minor points:
(1) The generated RNA-seq data must be deposit in a public domain, such as GEO.
Author Response
We would like to thank the reviewer for his/her feedback.
We have amended the text by improving background, methods, discussion and references, as suggested by all the reviewers.
Below you can find the point-by-point reply.
Major comments:
[1] “2.4. Viral genome amplification”. As far as what is written in this subsection, the authors did NOT treat RNA with DNase I to digest the genomic DNA in prior to reverse transcription. Although influenza virus is RNA virus, the rationale for not treating with DNase I must be stated clearly.
We have detailed it in the Experimental design (page 3 lines 113-115).
[2] “Geneious software 143 v. R11 (Biomatter, New Zealand; available at: https://www.geneious.com/)” The authors used the commercial available software product to analyze the generated data. Unless this product is freely available to the academic community, the authors must provide the computational pipeline using other freely available software products (i.e., without license fees) as it is not fair for others without access to this commercial software product, especially the authors clearly states as follow in the abstract: “The total turnaround 33 time per set of 20 samples was 4 working days, including sequence analysis.”
For this study we have planned to use the Geneious software, which is specific for the Illumina platform. This platform is used by many labs running NGS, therefore many users should have this analysis software available. However, in the attempt to address your suggestion, we have described the main steps of our computational pipeline in the text (page 9 lines 401-409) to allow other users to reproduce our analysis.
[3] “The study samples (N=80) had viral loads ranging from 104 to 1010 genome cop-157 ies/ml; all of which were successfully amplified to obtain the entire IAV genome; all puri-158 fied DNA was of a suitable quality and concentration (at least 0.2 ng/μl) to be sequenced 159 by NGS.” The authors must provide a table summarizing the patients’ information, such as age, gender, etc.
We have omitted to add a table summarizing the patients’ information, as long as in the current protocol format we believe that the patient’s information does not add value to the interpretation of the results. Of course, the reader who wants to have a look into these data or to get specific and technical explanations concerning the protocol can contact the corresponding author at any time.
Minor points:
(1) The generated RNA-seq data must be deposit in a public domain, such as GEO.
We are planning to deposit the study sequences in the influenza virus repository GISAID.
Round 3
Reviewer 3 Report
[3] “The study samples (N=80) had viral loads ranging from 104 to 1010 genome cop-157 ies/ml; all of which were successfully amplified to obtain the entire IAV genome; all puri-158 fied DNA was of a suitable quality and concentration (at least 0.2 ng/μl) to be sequenced 159 by NGS.” The authors must provide a table summarizing the patients’ information, such as age, gender, etc.
We have omitted to add a table summarizing the patients’ information, as long as in the current protocol format we believe that the patient’s information does not add value to the interpretation of the results. Of course, the reader who wants to have a look into these data or to get specific and technical explanations concerning the protocol can contact the corresponding author at any time.
=> The patient information must be provided as Supplementary Data.
Minor points:
(1) The generated RNA-seq data must be deposit in a public domain, such as GEO.
We are planning to deposit the study sequences in the influenza virus repository GISAID.
=> The data must be deposited in a public database and provide the accession number.
Author Response
We would like to thank the reviewer for his/her feedback.
Please, find below the point-by-point reply.
Major comments:
[3] “The study samples (N=80) had viral loads ranging from 104 to 1010 genome cop-157 ies/ml; all of which were successfully amplified to obtain the entire IAV genome; all puri-158 fied DNA was of a suitable quality and concentration (at least 0.2 ng/μl) to be sequenced 159 by NGS.” The authors must provide a table summarizing the patients’ information, such as age, gender, etc.
We have omitted to add a table summarizing the patients’ information, as long as in the current protocol format we believe that the patient’s information does not add value to the interpretation of the results. Of course, the reader who wants to have a look into these data or to get specific and technical explanations concerning the protocol can contact the corresponding author at any time.
=> The patient information must be provided as Supplementary Data.
We have detailed it in the supplementary table 1.
Minor points:
(1) The generated RNA-seq data must be deposit in a public domain, such as GEO.
We are planning to deposit the study sequences in the influenza virus repository GISAID.
=> The data must be deposited in a public database and provide the accession number.
The following acc. n. EPI-ISL14307235-250, 256, 261, 267-284, 287-316, 319-332 have been provided.
Round 4
Reviewer 3 Report
I have no further comment to make.
Author Response

(The authors gave the same response as above.)
